# Initial Efficacy of the COVID-19 mRNA Vaccine Booster and Subsequent Breakthrough Omicron Variant Infection in Patients with B-Cell Non-Hodgkin’s Lymphoma: A Single-Center Cohort Study

**DOI:** 10.3390/v16030328

**Published:** 2024-02-21

**Authors:** Makoto Saito, Akio Mori, Takashi Ishio, Mirei Kobayashi, Shihori Tsukamoto, Sayaka Kajikawa, Emi Yokoyama, Minoru Kanaya, Koh Izumiyama, Haruna Muraki, Masanobu Morioka, Takeshi Kondo

**Affiliations:** 1Blood Disorders Center, Aiiku Hospital, Sapporo 064-0804, Hokkaido, Japan; 2Division of Laboratory, Aiiku Hospital, Sapporo 064-0804, Hokkaido, Japan

**Keywords:** B-cell non-Hodgkin’s lymphoma, COVID-19 mRNA vaccine, booster effect, breakthrough infection, Omicron variant

## Abstract

It has been suggested that the effect of coronavirus disease 2019 (COVID-19) booster vaccination in patients with B-cell non-Hodgkin’s lymphoma (B-NHL) is inferior to that in healthy individuals. However, differences according to histological subtype or treatment status are unclear. In addition, there has been less research on patients who subsequently develop breakthrough infections. We investigated the effects of the first COVID-19 booster vaccination for patients with B-NHL and the clinical features of breakthrough infections in the Omicron variant era. In this study, B-NHL was classified into two histological subtypes: aggressive lymphoma and indolent lymphoma. Next, patients were subdivided according to treatment with anticancer drugs at the start of the first vaccination. We also examined the clinical characteristics and outcomes of patients who had breakthrough infections after a booster vaccination. The booster effect of the COVID-19 mRNA vaccine in patients with B-NHL varied considerably depending on treatment status at the initial vaccination. In the patient group at more than 1 year after the last anticancer drug treatment, regardless of the histological subtype, the booster effect was comparable to that in the healthy control group. In contrast, the booster effect was significantly poorer in the other patient groups. However, of the 213 patients who received the booster vaccine, 22 patients (10.3%) were infected with COVID-19, and 18 patients (81.8%) had mild disease; these cases included the patients who remained seronegative. Thus, we believe that booster vaccinations may help in reducing the severity of Omicron variant COVID-19 infection in patients with B-NHL.

## 1. Introduction

Approximately 4 years have passed since coronavirus disease 2019 (COVID-19) was declared a pandemic. Vaccination for severe acute respiratory syndrome coronavirus 2 (SARS-CoV-2) is thought to be effective at suppressing the spread of this infection and preventing its severity [1,2]. The number of new patients and severe cases has steadily decreased, and the World Health Organization has declared the end of the emergency state [3]. However, compared to healthy controls, B-cell non-Hodgkin’s lymphoma (B-NHL) patients reportedly respond poorly to the COVID-19 vaccine [4,5,6,7,8]. Unlike in the general healthy population, patients with hematological malignancies (especially B-NHL) are thought to be more susceptible to severe infections with SARS-CoV-2 [9]. COVID-19 keeps representing a threat in terms of clinical outcome for B-NHL patients. Also, from the time that the Omicron strain swept Japan in January 2022, it is important to know the outcomes for B-NHL patients who contract this emerging infectious disease.

In Japan, two doses of the mRNA vaccine were administered at 3-week intervals (BNT162b2 [Pfizer-BioNTech]) or 4-week intervals (mRNA-1273 [Moderna]), followed by an initial booster (third total dose) at a 5-month or longer interval. A fourth dose is mainly recommended for immunosuppressed individuals, and vaccinations are administered up to the seventh dose. In the current study, we evaluated the serological responses to three doses of the COVID-19 mRNA vaccine (the first booster in Japan) in patients with B-NHL.

Many B-NHL patients who received anti-CD20 monoclonal antibody therapy had reduced seroconversion after COVID-19 vaccination [10,11,12]. It has been suggested that the effect of booster vaccination is inferior to that of vaccination in healthy individuals [13,14,15]. In previous studies, differences according to histological B-NHL subtype and treatment status for lymphoma at the start of vaccination were not clear. In addition, Lee LYW et al. reported that the risk of SARS-CoV-2 breakthrough infection and hospitalization increases when antibody titers decrease below 5000 U/mL [16].

The clinical features of patients with B-NHL who develop breakthrough infections after a booster vaccination are not well understood. A recent report based on the European Society of Hematology registry included 1548 patients (1181 [76%] had lymphoid malignancies) and showed significantly lower 30-day mortality rates among patients with breakthrough COVID-19 infections after a booster vaccination (*n* = 143, 9%) than in the pre-vaccine era (31%) [17]. In this study, we clarified the effects of the first COVID-19 booster vaccination in patients with B-NHL during follow-up in our department and examined the clinical features of patients who subsequently developed breakthrough Omicron variant infections.

## 2. Materials and Methods

### 2.1. Study Design and Data Collection

In this prospective observational study, the subjects included patients with B-NHL who were followed up in our department and who received three doses of the COVID-19 mRNA vaccine from 17 August 2021 to 30 September 2022 (cross-vaccination was allowed for the third dose). In addition, healthy controls (HCs) were recruited from health care workers older than 50 years of age who were working at our hospital, had no history of hematological disease or COVID-19 infection, and received three vaccine doses during the same time period. Each serum sample was analyzed for anti-SARS-CoV-2 spike (S) antibodies by using the Elecsys^®^ Anti-SARS-CoV-2S Immunoassay (Roche Diagnostics, Basel, Switzerland). An antibody titer ≤ 0.8 U/mL was defined as negative, and an antibody titer > 0.8 U/mL was defined as positive according to the report by Kageyama et al. [18]. A change from seronegative to seropositive was defined as seroconversion.

Upon comparing the antibody titer (pre-value) 6 months after the second vaccination and the antibody titer (post-value) after the third vaccination, a booster effect was defined as inducing seropositivity after the third inoculation and increasing the antibody titer to more than twice the pre-value (according to the method described in the review by Mai AS et al. [14]). As mentioned earlier, it was also noted whether the antibody titer had increased to 5000 U/mL or higher. When the antibody titer was less than 0.4 U/mL or was 25,000 U/mL or more, the values were designated 0.4 U/mL and 25,000 U/mL, respectively, for further calculations. Antibody titers are displayed as common logarithms. Patients who became infected with SARS-CoV-2 before the third vaccine dose or whose pre- or post-antibody titers were not examined (despite receiving three doses) were excluded from this study. In this study, no patients received the Omicron strain-specific vaccine or the anti-SARS-CoV-2 monoclonal antibody tixagevimab/cilgavimab (AstraZeneca’s *Evusheld^®^*, formerly AZD7442).

### 2.2. Patients

First, patients with B-NHL were divided into two groups: an aggressive lymphoma (diffuse large B-cell lymphoma [DLBCL], intravascular lymphoma [IVL], primary central nervous system lymphoma, and primary mediastinal large B-cell lymphoma, among other conditions) group and an indolent lymphoma (follicular lymphoma [FL], chronic lymphocytic leukemia/small lymphocytic lymphoma [CLL/SLL], lymphoplasmacytic lymphoma, marginal zone lymphoma, and mantle cell lymphoma [MCL], among other conditions) group. If a transformation was suspected during follow-up, patients with lymphomas in which morphologically clear FL or mucosa-associated lymphoid tissue (MALT) lymphoma components were still present were classified into the indolent lymphoma group.

Second, depending on anticancer drug treatment status at the time of the first vaccination, the patients were subdivided into (1) patients at more than 1 year after the completion of treatment with anticancer drugs (off treatment for >1 year group); (2) patients at less than 1 year after completion of treatment with anticancer drugs (off treatment for <1 year group); (3) patients who started vaccination during treatment with anticancer drugs (during treatment group); and (4) patients who received two vaccinations before treatment with anticancer drugs (before treatment group). Patients with no history of treatment with anticancer drugs (treatment-naïve group) were also included in this study.

In addition, we examined the clinical characteristics and outcomes of patients who had breakthrough infections after a booster vaccination from February 2022 to January 2023. Herein, the severity of COVID-19 was classified according to Japan’s “COVID-19 Medical Treatment Guide, Version 9.0 (10 February 2023)” as follows [19]: mild—mild respiratory symptoms and no pneumonia findings; moderate-I—pneumonia findings but no respiratory failure; moderate-II—respiratory failure requiring supplemental oxygen; and severe—requires admission to the ICU or ventilator therapy.

### 2.3. Statistical Analysis

Median antibody titers were compared by using the Mann–Whitney U test. The Spearman’s rank correlation coefficient was used to assess the relationship between two variables. A two-tailed test with *p* < 0.05 was considered to indicate statistical significance. All of the statistical analyses were performed by using an EZR (Jichi Medical School, Saitama, Japan) [20], Version 2.3-0.

## 3. Results

### 3.1. Study Population

In the present study, a total of 241 patients with B-NHL were initially enrolled, and 28 patients dropped out before the third vaccination due to patient death (9), refusal of the third vaccination (8), discontinuation of follow-up (6), and COVID-19 infection during the course (5) (Figure 1). Afterward, among the 213 patients who received the third vaccine dose, 27 patients with antibody titers that were not measured before or after the third vaccination were also excluded. In total, 186 patients with B-NHL (median age: 71 [38–92]
years, 103 males, 83 females) and 26 HCs (median age: 55 [50–72] years, 10 males, 16 females)
were included in this study.
The median time interval from the second to third vaccination was 220 (175–371) days (at that time, in Japan, the rule was that the interval between the second and third doses of the COVID-19 vaccine should be “at least 5 months”). These 186 patients were subdivided according to the process described in Section 2.2. All 12 treatment-naïve patients were in the indolent lymphoma group.

### 3.2. Clinical Characteristics of Each Disease Group

In the aggressive lymphoma group (Table 1), 90 patients (median age: 71 [39–92] years, 47 males, 43 females) were included, 82 of whom had DLBCL and 72 of whom were treated with the R-CHOP (rituximab-cyclophosphamide, daunorubicin, vincristine, and prednisolone) (or similar) regimen. In the indolent lymphoma group (Table 2), there were 96 patients (median age: 69 [38–87] years, 56 males, 40 females), including 12 patients with no history of treatment with anticancer drugs. FL accounted for 61 patients, and R (or obinutuzumab, G)-CHOP (or similar) was the most common treatment (37 patients), followed by R (G)-B (bendamustine) or R-BAC (rituximab-bendamustine and cytarabine for MCL) (30 patients). Of the nine patients with CLL/SLL, three patients were treatment-naïve (“watch”), and four patients received Bruton’s tyrosine kinase (BTK) inhibitors.

### 3.3. Booster Effects after the Third Vaccination (Table 3)

#### 3.3.1. Healthy Control Group and Treatment-Naïve Group (Figure 2)

HCs (*n* = 26) and treatment-naïve indolent lymphoma patients (*n* = 12) had a seroconversion rate of 100% after the second vaccination. After the third inoculation, the antibody titers increased from 582.5 U/mL and 939.0 U/mL to 15,324.0 U/mL and 15,459.5 U/mL, respectively, thus indicating a booster effect. Twenty-four (92.3%) HCs and eleven patients (91.7%) in the treatment-naïve group had antibody titers elevated to ≥5000 U/mL.

**Table 3 viruses-16-00328-t003:** Booster effects (antibody titers) after the third vaccination for each subclassified disease state.

Histological Subtype		Aggressive Lymphoma				Indolent Lymphoma				
Treatment Status	HCs	Off >1 year	Off <1 year	During	Before	Naïve	Off >1 year	Off <1 year	During	Before
Number	26	45	28	9	8	12	40	18	17	9
Sex (M/F)	M10/F16	M24/F21	M16/F12	M3/F6	M4/F4	M9/F3	M22/F18	M10/F8	M11/F6	M4/F5
Age	55	69	72.5	74	77	68	68.5	65.5	68	73
Pre-value (U/mL)	582.5	680	<0.40	<0.40	100	939	434	<0.40	<0.40	28.1
Post-value (U/mL)	15,324	20,419	109.45	<0.40	41.85	15,459.5	17,071	8.49	<0.40	21.8
Seroconversion rate (%)	(-/0)	0 (0/1)	68.4 (13/19)	28.6 (2/7)	(-/0)	(-/0)	50 (1/2)	42.9 (6/14)	12.5 (2/16)	50 (1/2)
Booster effect (%)	100	97.8	78.6	11.1	12.5	100	97.5	50	17.6	22.2
Antibody titer > 5000 (%)	24 (92.3)	42 (93.3)	8 (28.6)	0 (0)	0 (0)	11 (91.7)	32 (80)	2 (11.1)	0 (0)	0 (0)

HCs: Healthy Controls.

**Figure 2 viruses-16-00328-f002:**
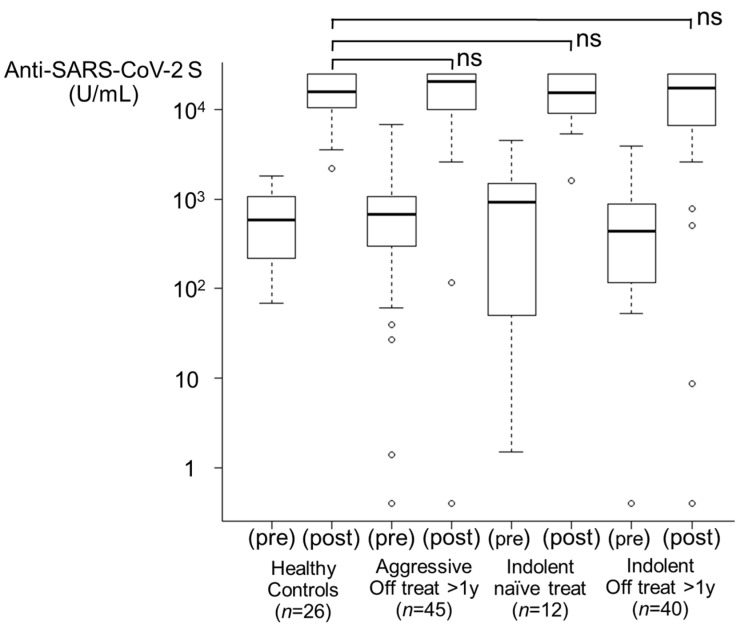
Booster effects after the third vaccination. From left to right, healthy control group (*n* = 26), aggressive lymphoma with treatment off for >1 year group (*n* = 45), indolent lymphoma with treatment-naïve group (*n* = 12), and indolent lymphoma with treatment off for >1 year group (*n* = 40) are shown. Regardless of the histological subtype including treatment-naïve group, the booster effect was comparable to that in the healthy control group. ns, not significant.

#### 3.3.2. Off Treatment for >1 Year Group (Figure 2)

Among the patients in the aggressive lymphoma group (*n* = 45), 44 patients had seropositive results after the second vaccination, and a booster effect after the third vaccination was observed (97.8%). Among the patients in the indolent lymphoma group (*n* = 40), 38 patients were already seropositive after the second dose, and a booster effect was shown in 39 patients (97.5%). The antibody titers increased from 680.0 U/mL and 434.0 U/mL to 20,419 U/mL and 17,071 U/mL, respectively. Forty-two patients (93.3%) in the aggressive lymphoma group and thirty-two patients (80.0%) in the indolent lymphoma group had antibody titers elevated to ≥5000 U/mL.

#### 3.3.3. Off Treatment for <1 Year Group (Figure 3A)

In the aggressive lymphoma group (*n* = 28), nine patients (32.1%) were seropositive after the second vaccination. After 3 doses, 13 patients exhibited successful seroconversion, and 22 patients (78.6%) exhibited a booster effect (median antibody titer of 207.5 U/mL, 1/74 HCs, *p* = 2.34 × 10^−6^). In the indolent lymphoma group (*n* = 18), four patients (22.2%) were seropositive after two doses of the vaccine. After the third dose, six patients exhibited successful seroconversion, and a booster effect was observed in nine patients (50.0%) (median antibody titer of 74.6 U/mL, 1/205 HCs, *p* = 8.01 × 10^−8^). None of these nine patients received ≥ six cycles of bendamustine. The antibody titers of both the aggressive lymphoma group and the indolent lymphoma group were significantly lower than those of the HCs. In addition, a lower titer in the indolent lymphoma group compared to the aggressive lymphoma group was observed, although statistical significance was not reached (*p* = 0.0603). Eight patients (28.6%) in the aggressive lymphoma group and two patients (11.1%) in the indolent lymphoma group had antibody titers elevated to ≥5000 U/mL.

#### 3.3.4. During Treatment Group (Figure 3B)

Five of the nine patients (55.6%) in the aggressive lymphoma group and thirteen of the seventeen patients (76.5%) in the indolent lymphoma group were persistently seronegative from 3 months after the second vaccination to even after the third dose. A booster effect was observed in only one patient in the aggressive lymphoma group (11.1%, *p* = 9.45 × 10^−6^ for HCs) and three patients in the indolent lymphoma group (17.6%, *p* = 2.91 × 10^−8^ for HCs). The antibody titers of both the aggressive lymphoma group and the indolent lymphoma group were significantly lower than those of the HCs. In the indolent lymphoma group, six patients were treated with BTK inhibitors (mainly ibrutinib), only one of whom had a booster effect. In both groups, none of the patients had antibody titers elevated to ≥5000 U/mL.

#### 3.3.5. Before Treatment Group (Figure 3C)

A total of 17 patients (8 in the aggressive lymphoma group and 9 in the indolent lymphoma group) had a median time of 110 (range: 8–166) days from the completion of two vaccination doses to treatment with anticancer drugs. In 14 patients (82.4%), the antibody titer was lower after 3 doses (41.85 U/mL and 21.8 U/mL in the aggressive lymphoma group and the indolent lymphoma group, respectively) than after 2 doses (100.0 U/mL and 28.1 U/mL, respectively). A booster effect was observed in only one patient in the aggressive lymphoma group (12.5%, *p* = 2.34 × 10^−5^ for HCs) and two patients in the indolent lymphoma group (22.2%, *p* = 1.38 × 10^−5^ for HCs). However, one of the latter patients had hairy cell leukemia and remained mainly under observation (“watch”) except for 1 week of treatment with cladribine alone. The antibody titers of both the aggressive lymphoma group and the indolent lymphoma group were significantly lower than those of the HCs. In both groups, none of the patients had antibody titers elevated to ≥5000 U/mL.

**Figure 3 viruses-16-00328-f003:**
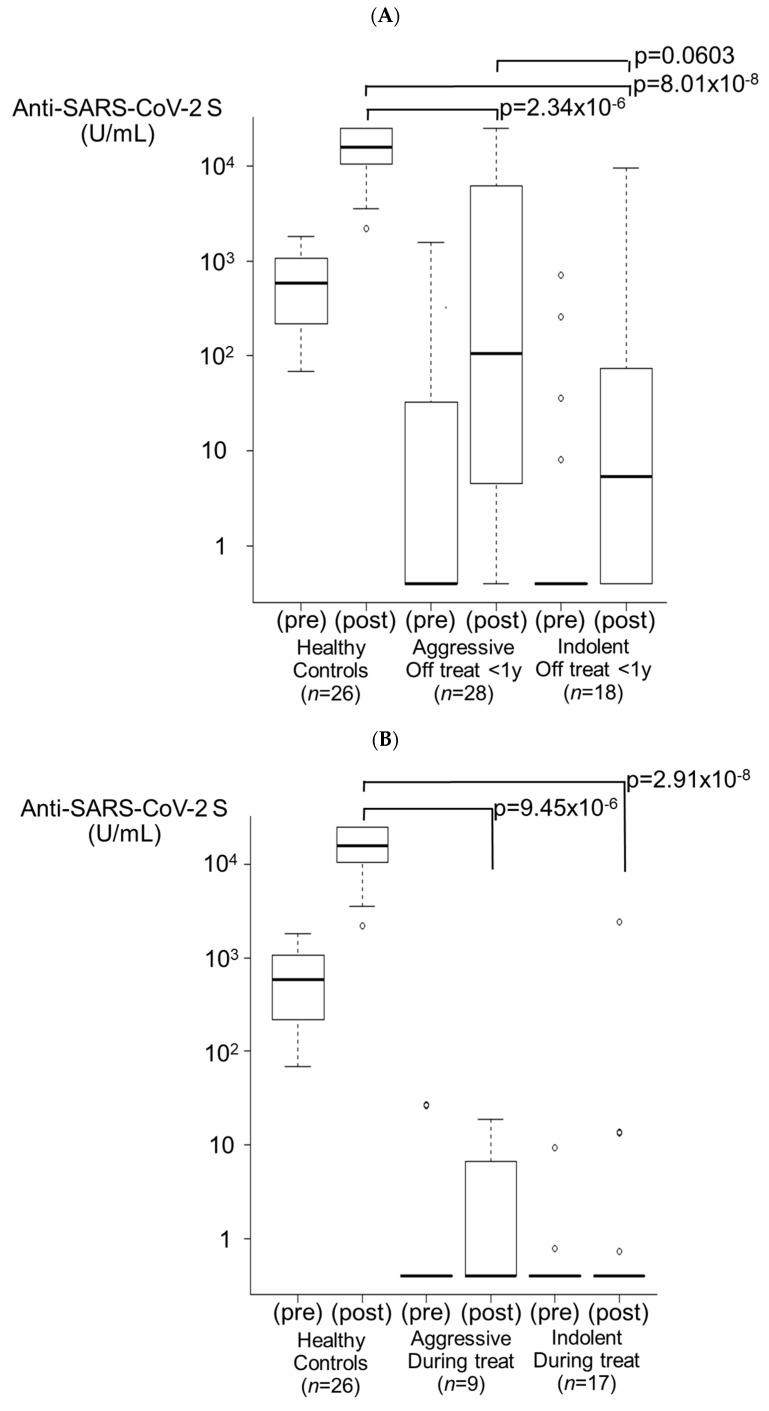
Booster effects after the third vaccination. (**A**) Treatment off for <1 year group. In the aggressive lymphoma group, 78.6% of patients (*n* = 28) exhibited a booster effect (*p* = 2.34 × 10^−6^). In the indolent lymphoma group, 50.0% of patients (*n* = 18) exhibited a booster effect (*p* = 8.01 × 10^−8^). The antibody titer in the indolent lymphoma group tended to be lower than that in the aggressive lymphoma group (*p* = 0.0603). (**B**) During treatment group. In the aggressive lymphoma group, 11.1% of patients (*n* = 9) exhibited a booster effect (*p* = 9.45 × 10^−6^). In the indolent lymphoma group, 17.6% of patients (*n* = 17) exhibited a booster effect (*p* = 2.91 × 10^−8^). (**C**) Before treatment group. In the aggressive lymphoma group, 12.5% of patients (*n* = 8) exhibited a booster effect (*p* = 2.34 × 10^−5^). In the indolent lymphoma group, 22.2% of patients (*n* = 9) exhibited a booster effect (*p* = 1.38 × 10^−5^).

### 3.4. Breakthrough Infection after Booster Vaccination (Table 4A,B)

Of the 213 patients who were enrolled in this study who received the first booster dose, 22 patients (10.3%, median age: 68.5 [44–91] years, 12 males and 10 females) developed a breakthrough infection with SARS-CoV-2 after the third or fourth dose. The clinical characteristics of the patients with breakthrough COVID-19 infections are shown in Table 4A,B. Most of the patients (18, 81.8%) had mild disease. Seventeen patients had community-acquired infections, and five patients had nosocomial infections. Among the latter patients, antiviral treatment (mainly molnupiravir) was administered regardless of the severity of symptoms. The underlying disease was aggressive lymphoma (all cases were DLBCL) in seven patients and indolent lymphoma in fifteen patients. The median time from the last vaccination to COVID-19 infection was 158 (range: 48–195) days in the aggressive lymphoma (DLBCL) group and 75 (range: 5–306) days in the indolent lymphoma group. All seven patients with DLBCL demonstrated a booster effect (for a median antibody titer of 12,142 [1268–>25,000] U/mL) due to the vaccine, regardless of lymphoma treatment status at the start of vaccination, and the symptoms of COVID-19 infection were mild (Table 4A). Among them, five patients with community-acquired infections (Cases 1–4, 6) received only usual care for a flu-like cold. Two patients (Cases 5 and 7) who were infected in the hospital were treated with molnupiravir, and chemotherapy for DLBCL was resumed after testing negative for SARS-CoV-2. In the end, both patients achieved complete remission.

In contrast, of the 15 indolent lymphoma patients with breakthrough infections, 11 patients (73.3%) had mild disease, 2 patients (13.3%) had moderate-I disease, 1 patient (6.7%) had moderate-II disease, and 1 patient (6.7%) died (Table 4B). Seven patients were classified as the ‘off treatment for >1 year group’, and the median value of the antibody titer was 56 (<0.4–1–6384) U/mL. Eight patients were seropositive for COVID-19 before infection, and five patients had mild disease and received usual care alone. Seven patients (Cases 13, 15, 17–19, 21, and 22), including four patients on maintenance treatment with obinutuzumab, were seronegative before infection despite ≥ three vaccinations; moreover, except for Case 19, six patients remained seronegative even after infection. Four of them received molnupiravir after infection, and three patients received usual care; however, two of the latter patients (Cases 13 and 22) progressed to moderate-I disease (a pneumonia shadow was revealed on CT more than 1 month after the onset of infection). The patient (designated Case 19) was infected during the period of maintenance treatment with obinutuzumab; moreover, despite not being persistently positive, repeated remissions (COVID-19-negative) and relapses (COVID-19-positive) occurred seven times for nearly 1 year. The patient was frequently hospitalized with pneumonia. Additionally, the patient (designated as Case 14) was hospitalized for treatment of recurrent MCL and was unfortunately infected before starting chemotherapy. Initially, the infection was assessed to be mild, and his condition did not seem severe for 1 week after infection. However, as the patient became persistently infected with COVID-19 despite the initial administration of remdesivir and sotrovimab, the lymphoma progressed further, and he died 25 days after SARS-CoV-2 infection with severe respiratory and renal failure. In addition to being infected with COVID-19, it was also thought that the death was due to the progression of lymphoma.

**Table 4 viruses-16-00328-t004:** Clinical characteristics of the patients with breakthrough COVID-19 infection after booster vaccination ((**A**): aggressive lymphoma, (**B**): indolent lymphoma).

(**A**)											
	**Age**	**Sex**	**Histology**	**Last Treatment** **Regimen**	**Lymphoma Treatment Status at the Start of Vaccination**	**Number of** **Vaccinations** **(Time to Onset)**	**Pre-infection Antibody Titer (U/mL)**	**Severity**	**Infectious Status**	**COVID-19 Treatment**	**Post-Infection Antibody Titer (U/mL)**
1	70	M	DLBCL	R-CHOP	Off >1 y	3 (163)	>25,000	mild	community infection	Usual care	>25,000
2	65	F	DLBCL	R-EPOCH + MTX	Off >1 y	3 (99)	24,148	mild	community infection	Usual care	>25,000
3	69	M	DLBCL	R-CHOP	Off >1 y	3 (158)	12,142	mild	community infection	Usual care	>25,000
4	54	F	DLBCL	R-CHOP	Off >1 y	3 (58)	3902	mild	community infection	Usual care	>25,000
5	78	F	DLBCL	Pola-BR	Off <1 y	4 (48)	5523	mild	nosocomial infection (under treatment-1 course)	Molnupiravir	1478
6	44	F	DLBCL	R-CHOP	Before	3 (184)	19,663	mild	community infection	Usual care	10,479
7	68	F	DLBCL	R-CHOP	Before	3 (195)	1268	mild	nosocomial infection (under treatment-5 courses)	Molnupiravir	1327
(**B**)											
	**Age**	**Sex**	**Histology**	**Last Treatment** **Regimen**	**Lymphoma Treatment Status at the Start of Vaccination**	**Number of** **Vaccinations** **(Time to Onset)**	**Pre-Infection Antibody Titer (U/mL)**	**Severity**	**Infectious Status**	**COVID-19 Treatment**	**Post-Infection Antibody Titer (U/mL)**
8	79	M	FL	R-CHOP	Off >1 y	3 (88)	16,384	mild	community infection	Usual care	>25,000
9	91	M	FL	RTX	Off >1 y	3 (19)	56	mild	nosocomial infection (other diseases)	Molnupiravir	>25,000
10	61	F	MALT	BR + R maintenance	Off >1 y	3 (5)	46.8	mild	community infection	Usual care	>25,000
11	76	F	FL	R-CHOP/R-F + R maintenance	Off >1 y	4 (114)	8833	mild	community infection	Usual care	>25,000
12	71	M	FL	BR	Off >1 y	4 (76)	1664	mild	community infection	Usual care	2581
13	58	M	FL	auto-SCT	Off >1 y	4 (60)	<0.4	moderate-1	community infection	Usual care	<0.4
14	71	M	MCL	BR	Off >1 y	4 (75)	38.6	death	nosocomial infection (before treatment)	Remdesivir, Sotrovimab	8318
15	51	F	FL	GB	Off <1 y	3 (81)	<0.4	mild	community infection	Molnupiravir	<0.4
16	75	M	FL	RTX	Off <1 y	4 (50)	848	mild (sequelae)	nosocomial infection (other diseases)	Molnupiravir	2029
17	51	M	FL	CAR-T	Off <1 y	4 (123)	0.53	mild	community infection	Molnupiravir	<0.4
18	56	M	FL	G-CHOP + G maintenance	During	3 (306)	<0.4	mild	community infection	Usual care	<0.4
19	60	F	FL	GB + G maintenance	During	3 (34)	<0.4	moderate-2	community infection	Molnupiravir, Sotrovimab, Nirmatrelvir/Ritonavir, Remdesivir	9366
20	67	M	FL	BR + R maintenance	During	4 (55)	14,685	mild	community infection	Usual care	>25,000
21	73	M	FL	GB + G maintenance	During	4 (61)	<0.4	mild	community infection	Molnupiravir	<0.4
22	70	F	FL	GB + G maintenance	During	4 (191)	<0.4	moderate-1	community infection	Usual care	<0.4

y: year, m: month, DLBCL: diffuse large B-cell lymphoma, FL: follicular lymphoma, MALT: mucosa-associated lymphoid tissue lymphoma, MCL: mantle cell lymphoma, R (G)-CHOP: rituximab (obinutuzumab)-cyclophosphamide, daunorubicin, vincristine and prednisolone, EPOCH: etoposide, prednisone, vincristine, cyclophosphamide and doxorubicin, MTX: methotrexate, Pola-BR: Polatuzumab vedotin combined with bendamustine and rituximab, R-F: rituximab and Fludarabine, RTX: rituximab, R(G)-B: rituximab (obinutuzumab)-bendamustine, CAR-T: chimeric antigen receptor-T cell therapy, SCT: stem cell transplantation.

## 4. Discussion

In our current study, there was no difference between HCs (*n* = 26) and treatment-naïve indolent lymphoma patients (*n* = 12) because the number of cases was too small; however, according to a recently published review [21], among 1135 patients with untreated hematological malignancies, primarily consisting of patients with CLL or indolent B-NHL, the seroconversion rate was suppressed to 80.4%, suggesting an inherent impairment of immunogenicity. In patients in both the aggressive lymphoma group and the indolent lymphoma group who were vaccinated more than 1 year after the final anticancer treatment, the booster effect was equal to that in the HCs. Conversely, in the group of patients who were vaccinated less than 1 year after the completion of treatment with anticancer drugs, the booster effect was significantly poorer than that in the group of HCs. In previous reports, among CLL/SLL or B-NHL patients who failed to achieve a humoral response after two standard vaccinations, the response rate to the third vaccination was in the 20% range (23.8%, 29.5%, respectively) [11,12]. This is likely because anti-CD20 monoclonal antibodies target memory B-cells and suppress humoral immunity for more than 6 months [12,22], and recovery from peripheral blood B-cell depletion has been shown to take approximately 1 year [22,23]. Especially in the indolent lymphoma group, the decrease in the booster effect was more obvious. One of the reasons may be due to the treatment with bendamustine and ibrutinib [24,25,26], which were rarely used for DLBCL at the time of this study. Additionally, it has been reported that patients with indolent B-NHL have longer seroconversion times after vaccination than patients with aggressive B-NHL [7]. Furthermore, the booster effect was significantly poorer in the group of patients who started vaccination during treatment with anticancer drugs, such as BTK inhibitors or maintenance therapy with anti-CD20 antibody drugs in the indolent lymphoma group. In the group of patients who had been vaccinated twice before anticancer drug treatment, the antibody titer decreased even after the third vaccination, thus indicating that the antibody titer decreased after chemotherapy. When treating patients with B-NHL, the timing of booster vaccinations should be carefully considered.

Since January 2022, Omicron has been the most frequently detected variant strain of SARS-CoV-2 in both cities of Hokkaido and also throughout Japan [27]. The prevalence of anti-N antibodies induced by SARS-CoV-2 infection increased from 24.1% to 37.6% in the general population (approximately 70% had received their third vaccination during the same time period) in cities of Hokkaido from November 2022 to February 2023 [28]. In the overall cohort of 213 B-NHL patients who received a third or fourth booster vaccine, 22 patients (10.3%) subsequently developed breakthrough SARS-CoV-2 infection from February 2022 to January 2023. Although a simple comparison was not possible, booster vaccinations may be helpful for preventing infection in B-NHL patients.

The Omicron variant is known to be more transmissible but milder in the general healthy population compared to past strains, such as the Delta strain [29,30]. Only a few studies have reported on the impact of the Omicron variant of SARS-CoV-2 on clinical outcomes among patients with hematological malignancies. A report based on the registry of the European Society of Hematology reported a mortality rate of 7.9% among patients with hematological malignancies infected with the Omicron variant, which is similar to that reported for patients infected with other variants [17]. Zhu et al. reported that the severity of infection caused by Omicron strains in patients with hematological malignancies was 5.3%, and the mortality rate was 2.2% [31]. According to our current study, in the entire cohort of 213 patients, 22 patients developed breakthrough SARS-CoV-2 infection, and the majority of patients (18, 81.8%), including those patients infected without booster effects and those who did not receive antiviral drugs after infection, had mild symptoms similar to a flu-like cold and recovered within a few days. Only two patients (9.1% of infected patients, 0.9% of the entire cohort) had severe conditions requiring respiratory management (moderate-II to severe according to the Japanese classification), and the mortality rate was 0.5% in this cohort. This rate is lower than the rates of severe disease and mortality mentioned above, and it is thought that booster vaccinations contribute to suppressing the severity of the disease. One reason for this difference is thought to be that all seven infected patients with DLBCL had high antibody titers (≥5000 U/mL in five patients) after booster vaccination. In contrast, of the 15 patients with indolent lymphoma, 11 (73.3%) had mild symptoms, but 4 (26.7%) had more severe symptoms. This is because they are immunogenic disorders that are specific to indolent lymphoma [21]. In two patients with pneumonia equivalent to moderate (-I/II) disease, though vaccination was started during maintenance therapy with obinutuzumab, they developed breakthrough infections while their antibody tests were negative. The risk of COVID-19 infection associated with obinutuzumab administration has recently been discussed [32], and active treatment with this drug requires careful consideration. Furthermore, four of the six infected patients with indolent lymphoma who remained seronegative even after infection had mild disease. We speculate that repeated vaccination stimulates specific T-cell responses and activates cellular immunity, as reported by Uaprasert N et al. [21] and Haggenburg S et al. [33]. However, the Omicron strain is weakly virulent, so it cannot be ruled out that it may have been in a state of “immunological tolerance”. Through our current research, the reason that most patients with breakthrough infection did not develop severe symptoms is thought to be due to both the effectiveness of the COVID-19 booster vaccination and the low toxicity of the Omicron variant.

This study presented some limitations. First, patients were subdivided into many groups, and the number of patients in each group varied. Second, the details of each patient’s humoral and cellular immunity were not investigated. Additionally, it was not possible to compare the results with those of a group of unvaccinated patients. In the pre-Omicron era, Chien KS et al. showed that, compared with unvaccinated patients, vaccinated patients with hematologic malignancies were associated with a reduced risk of hospitalization due to COVID-19 infection [34]. Data on COVID-19 Omicron variant infections in patients with B-NHL are still lacking, and additional data with a larger patient pool are needed. In the future, we would like to investigate the actual situation of B-cell lymphoma cases with breakthrough infection, including patients who developed after this study, and re-evaluate the effectiveness of this vaccine.

## 5. Conclusions

The efficacy of the COVID-19 booster vaccine in patients with B-NHL varied significantly depending on the status of the final anticancer drug treatment. The booster effect in patients who started vaccination >1 year after completion of treatment (off treatment for >1 year) was equal to that in the HCs. In other patients (off treatment for <1 year, during treatment, and before treatment), regardless of the histological type, the booster effect was significantly poorer. As a result, approximately 10% of patients who received the third or fourth dose developed a breakthrough infection. More than 80% of patients had mild disease, even including patients who remained seronegative. In patients with B-NHL in the Omicron era, we believe that booster vaccination may help to reduce the severity of COVID-19 infection.

## Figures and Tables

**Figure 1 viruses-16-00328-f001:**
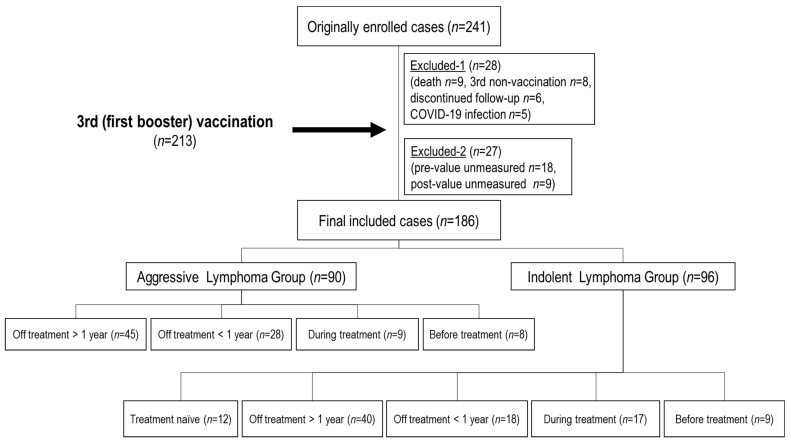
Patient flowchart. A total of 241 patients were initially enrolled. 28 patients dropped out before the third (initial booster) vaccination (9 died, 8 did not receive the third dose, 6 were discontinued from follow-up for lymphoma, and 5 were infected with SARS-CoV-2). 213 patients who received the third vaccine dose, 27 patients whose antibody titers were not measured before or after the third vaccination (18 pre- and 9 post-values) were also excluded. *n*: number.

**Table 1 viruses-16-00328-t001:** Clinical characteristics of patients with aggressive lymphoma.

**Age**	71 (39–92)	
**Sex**	M 47, F 43	
**Disease**		
DLBCL	82	
IVL	5	
PCNSL	1	
PML-BL	1	
Unclassifiable	1	
**Treatment Status**		
Off >1 year	45	
Off <1 year	28	
During	9	
Before	8	
**Treatment Regimen**		
R-CHOP like	72	
R + salvage	16	
SCT	7 (auto 6, allo 1)	
BR (/R monotherapy)	3	
**Vaccine subtype**	First/Second doses	Third dose
BNT162b2	80	58
mRNA-1273	10	32

DLBCL: diffuse large B-cell lymphoma, IVL: intravascular lymphoma, PCNSL: primary central nervous system lymphoma, PML-BL: primary mediastinal large B-cell lymphoma, R-CHOP: rituximab-cyclophosphamide, daunorubicin, vincristine and prednisolone, SCT: stem cell transplantation, B-R (rituximab-bendamustine).

**Table 2 viruses-16-00328-t002:** Clinical characteristics of patients with indolent lymphoma.

**Age**	69 (38–87)	
**Sex**	M 56, F 40	
**Disease**		
FL	61	
CLL/SLL *	9	
LPL	9	
MZL **	7	
MCL	7	
Others	3	
**Treatment Status**		
Naïve	12	
Off >1 year	40	
Off <1 year	18	
During	17	
Before	9	
**Treatment Regimen**		
R (G)-CHOP like	37	
R (G)-B/R-BAC	30	
R (G) monotherapy	8	
BTK inhibitors	7	
R-salvage	4	
SCT (auto-)	2	
CAR-T	1	
Others	5	
**Vaccine subtype**	First/Second doses	Third dose
BNT162b2	82	65
mRNA-1273	14	31

FL: follicular lymphoma, CLL/SLL: chronic lymphocytic leukemia/small lymphocytic lymphoma, LPL: lymphoplasmacytic lymphoma, MZL: marginal zone lymphoma, MCL: mantle cell lymphoma, R (G)-CHOP: rituximab (or obinutuzumab)-cyclophosphamide, daunorubicin, vincristine and prednisolone, R(G)-B: rituximab (or obinutuzumab)-bendamustine, R-BAC: rituximab-bendamustine and cytarabine, BTK: Bruton’s tyrosine kinase, SCT: stem cell transplantation, CAR-T: chimeric antigen receptor-T cell therapy. * including hairy cell leukemia, ** including mucosa-associated lymphoid tissue (MALT) lymphoma and splenic marginal zone lymphoma (SMZL).

## Data Availability

The data that support the findings of this study are available from the corresponding author upon reasonable request.

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
