# Peer review of "Initial Efficacy of the COVID-19 mRNA Vaccine Booster and Subsequent Breakthrough Omicron Variant Infection in Patients with B-Cell Non-Hodgkin’s Lymphoma: A Single-Center Cohort Study"

_viruses, 2024, doi:10.3390/v16030328_

Round 1
Reviewer 1 Report
Comments and Suggestions for Authors
This manuscript describes the efficacy of a third (booster) mRNA vaccination against COVID-19 in patients with B-cell Non-Hodgkin’s Lymphoma.
B-cells play a central role in the process of developing humoral (antibody-related) immunity after infection or vaccination. Most patients with B-cell malignancies require B-cell directed treatment known to significantly reduce the numbers of mature (CD-20 positive) B-cells and to interfere with antibody production. Thus, on one hand viral infections may be especially dangerous for these patients, but on the other hand, efficacy of vaccination is debatable.
Here, the authors present some interesting and at least in part previously unreported findings:
- A third (booster) vaccination can increase antibody titers and thus assumingly immunity against SARS-CoV-2 in patients with B-cell lymphoma.
- The booster effect is more or less the same in individuals with highly aggressive lymphoma and in those with indolent lymphoma.
- However, the humoral vaccination efficacy (measured as antibody titres) varies considerably due to treatment status: Patients who had completed B-cell directed anti-cancer treatment more than a year before the start (first dose) of vaccination responded to vaccination as well as healthy controls did. Patients who had either completed treatment less than a year before the start of vaccination, or who had been vaccinated during or before anti-cancer treatment, responded significantly poorer to vaccination, if at all.
- This observation is of relevance to those caring for patients with B-cell cancer, as it shows that vaccinations should probably be started or refreshed around one year after the end of treatment. While the authors have shown this für COVID-19 mRNA vaccinations only, there is no reason to assume that other vaccinations should follow other schedules. However, the finding may be specific for patients who receive B-cell directed therapy. (Similar studies in patients with other anti-cancer treatments might be interesting.)
- The interpretation of breakthrough infections in this patient cohort is more difficult to interpret. There were only few breakthrough infections (22 of 213 patients, i.e. about 10%), but if this relatively low rate is attributable to vaccination or if other factors (e.g., stricter isolation strategies in cancer patients) play a role cannot be told. Similarly, the severity of breakthrough infections is hard to interpret. While only two patients had severe conditions requiring respiratory management, and the mortality rate was 0.5% only, the study was not designed to analyse this further. Shifts in SARS-CoV-2 variants (from Delta to Omikron), treatment modalities of COVID-19 and other factors may have contributed to this outcome.
The authors present their study design and results clearly. The discussion is appropriate, highlighting both strengths and limitations adequately. The conclusions drawn are sound, especially that booster vaccinations more than a year after anti-B-cell treatment may help to reduce the burden of COVID-19 infections in patients with B-cell lymphoma. This is of practical relevance to all caregivers of such patients.
Author Response
Thank you very much for your high evaluation. I will send you the paper with some additions. I appreciate your continued support.
Reviewer 2 Report
Comments and Suggestions for Authors
A comprehensive overview of the objectives and background of the study is provided in the introduction.
The study design is clearly presented.
The serological analysis method is clarified in detail, however the choice of an antibody's titer cutoff (0.8 U/ml) should have a clear justification, particularly since the test insert (https://diagnostics.roche.com/global/en/products/params/elecsys-anti-sars-cov-2.html) uses a 1.0 = non-reactive cutoff. Give a succinct explanation or reference for this decision. Although an approach for dealing with extreme antibody titers (less than 0.4 U/mL or 25,000 U/mL) is described, the study would be more transparent if these values were justified.
This is an important reference that attests to the timing of the Omicron variant in Japan, especially since the study included the three doses of COVID-19 vaccine from 17 August 2021 to 30 September 2022. Moreover, the World Health Organization first reported the appearance of the Omicron variant (B. 1.1. 529) on 24 November 2021.
The statistical methods are adequately described, but I recommend specifying the version of EZR used.
The manuscript does not specify all the exclusion criteria mentioned in Figure 1.
It is important to justify the choice of the median time interval from the second to the third vaccination (175-371 days), considering the recommendations of the manufacturers of the SARS-CoV-2 vaccine.
The terms "First/Second" and "Third" in the columns entitled "Vaccine subtype" in Table 2 need to have their meanings made clearer. If they relate to doses, they should be mentioned specifically to prevent any misunderstandings.
In tables 4 A and 4 B it is necessary to explain the use of the asterisks.
Row 312- I suggest providing specific findings when referring to previous studies (e.g. references [11, 12])
Row 322- Provide extra information about the implications of timing of booster, especially for patients on BTK inhibitors or undergoing anti-CD20 antibody maintenance therapy, because the booster effect was significantly poorer (add p value).
It would be useful to consider how comorbidities and different treatment plans might affect the study results.
At the end of the discussion, suggestions for future research directions would be welcome.
In conclusion, the diversity of the population studied must be specified.
Explained the reason why, despite immunization, some patients remained seronegative. What effects does this have on how they defend against COVID-19?
Author Response
I have attached the file.
